# Multimodal Word Distributions

## Abstract

Word embeddings provide point representations of words containing useful semantic information. We introduce multimodal word distributions formed from Gaussian mixtures, for multiple word meanings, entailment, and rich uncertainty information. To learn these distributions, we propose an energy-based max-margin objective. We show that the resulting approach captures uniquely expressive semantic information, and outperforms alternatives, such as word2vec skip-grams, and Gaussian embeddings, on benchmark datasets such as word similarity and entailment.

## 1 Introduction

To model language, we must represent words. We can imagine representing every word with a binary one-hot vector corresponding to a dictionary position. But such a representation contains no valuable semantic information: distances between word vectors represent only differences in alphabetic ordering. Modern approaches, by contrast, learn to map words with similar meanings to nearby points in a vector space (Mikolov et al., 2013a), from large datasets such as Wikipedia. These learned word embeddings have become ubiquitous in predictive tasks.

Vilnis and McCallum (2014) recently proposed an alternative view, where words are represented by a whole probability distribution instead of a deterministic point vector. Specifically, they model each word by a Gaussian distribution, and learn its mean and covariance matrix from data. This approach generalizes any deterministic point embedding, which can be fully captured by the mean vector of the Gaussian distribution. Moreover, the full distribution provides much richer information than point estimates for characterizing words, representing probability mass and uncertainty across a set of semantics.

However, since a Gaussian distribution can have only one mode, the learned uncertainty in this representation can be overly diffuse for words with multiple distinct meanings (polysemys), in order for the model to assign *some* density to any plausible semantics (Vilnis and McCallum, 2014). Moreover, the mean of the Gaussian can be pulled in many opposing directions, leading to a biased distribution that centers its mass mostly around certain meaning while leaving the others not well represented.

In this paper, we propose to represent each word with an expressive multimodal distribution, for multiple distinct meanings, entailment, heavy tailed uncertainty, and enhanced interpretability. For example, one mode of the word 'bank' could overlap with distributions for words such as 'finance' and 'money', and another mode could overlap with the distributions for 'river' and 'creek'. It is our contention that such flexibility is critical for both qualitatively learning about the meanings of words, and for optimal performance on many predictive tasks.

In particular, we model each word with a mixture of Gaussians (Section 3.1). We learn all the parameters of this mixture model using a maximum margin energy-based ranking objective (Joachims, 2002; Vilnis and McCallum, 2014) (Section 3.3), where the energy function describes the affinity between a pair of words. For analytic tractability with Gaussian mixtures, we use the inner product between probability distributions in a Hilbert space, known as the expected likelihood kernel (Jebara et al., 2004), as our energy function (Section 3.4). Additionally, we propose transformations for numerical stability and initialization, resulting in a robust, straightforward, and scal-

able learning procedure, capable of training on a corpus with billions of words in days. We show that the model is able to automatically discover multiple meanings for words (Section 4.2), and significantly outperform other alternative methods across several tasks such as word similarity and entailment (Section 4.3, 4.4, 4.6).

## 2   Related Work

In the past decade, there has been an explosion of interest in word vector representations. word2vec, arguably the most popular word embedding, uses continuous bag of words and skip-gram models, as well as negative sampling for efficient conditional probability estimation (Mikolov et al., 2013a,b). Other popular approaches use feedforward (Bengio et al., 2003) and recurrent neural network language models (Mikolov et al., 2010, 2011; Collobert and Weston, 2008) to predict missing words in sentences, producing hidden layers that can act as word embeddings that encode semantic information. A different approach to learning word embeddings is through factorization of word co-occurrence matrices such as GloVe embeddings (Pennington et al., 2014). The matrix factorization approach has been shown to have an implicit connection with skip-gram and negative sampling Levy and Goldberg (2014). Bayesian matrix factorization where row and columns are modeled as Gaussians has been explored in Salakhutdinov and Mnih (2008) and provides a different probabilistic perspective of word embeddings.

In exciting recent work, Vilnis and McCallum (2014) propose a Gaussian distribution to model each word. Their approach is significantly more expressive than typical point embeddings, with the ability to represent concepts such as *entailment*, by having the distribution for one word (e.g. 'music') encompass the distributions for sets of related words ('jazz' and 'pop'). However, with a unimodal distribution, their approach cannot capture multiple distinct meanings, much like most deterministic approaches. A small body of work develops multiple deterministic embeddings that can capture polysemys, for example through a cluster centroid of context vectors (Huang et al., 2012), or an adapted skip-gram model with an EM algorithm to learn multiple latent representations per word Tian et al. (2014). Another related work by Nalisnick and Ravi (2015) models embeddings in

infinite-dimensional space where each embedding can gradually represent incremental word sense if complex meanings are observed.

Probabilistic word embeddings have only recently begun to be explored, and have so far shown great promise. In this paper, we propose, to the best of our knowledge, the first probabilistic word embedding that can capture multiple meanings. We use a Gaussian mixture model which allows for a highly expressive distributions over words. At the same time, we retain scalability and analytic tractability with an expected likelihood kernel energy function for training. The model and training procedure harmonize to learn descriptive representations of words, with superior performance on several benchmarks.

## 3   Methodology

In this section, we introduce our Gaussian mixture (GM) model for word representations, and present a training method to learn the parameters of the Gaussian mixture. This method uses an energy-based maximum margin objective, where we wish to maximize the similarity of distributions of nearby words in sentences. We propose an energy function that compliments the GM model by retaining analytic tractability. We also provide critical practical details for numerical stability and initialization.

### 3.1   Word Representation

We represent each word $w$ in a dictionary as a Gaussian mixture with $K$ components. Specifically, the distribution of $w$, $f_w$, is given by the density

$$f_w(\vec{x}) = \sum_{i=1}^{K} p_{w,i}\, \mathcal{N}\left[\vec{x}; \vec{\mu}_{w,i}, \Sigma_{w,i}\right] \qquad (1)$$

$$= \sum_{i=1}^{K} \frac{p_{w,i}}{\sqrt{2\pi|\Sigma_{w,i}|}} e^{-\frac{1}{2}(\vec{x}-\vec{\mu}_{w,i})^{\top}\Sigma_{w,i}^{-1}(\vec{x}-\vec{\mu}_{w,i})},$$

where $\sum_{i=1}^{K} p_{w,i} = 1$. The mean vectors $\vec{\mu}_{w,i}$ represent the location of the $i^{th}$ component of word $w$, and are akin to the point embeddings provided by popular approaches like word2vec. $p_{w,i}$ represents the component probability (mixture weight), and $\Sigma_{w,i}$ is the component covariance matrix, containing uncertainty information. Our goal is to learn all of the model parameters $\vec{\mu}_{w,i}, p_{w,i}, \Sigma_{w,i}$ from a corpus of natural sentences to extract semantic information of words. Each

Gaussian component's mean vector of word $w$ should be able to represent one of the word's distinct meanings. For instance, one component of a polysemous word such as rock should represent the meaning related to stone or pebbles, whereas another component should represent the meaning related to music such as jazz, pop. Figure 1 illustrates our word embedding model, and the difference between multimodal and unimodal representations, for words with multiple meanings.

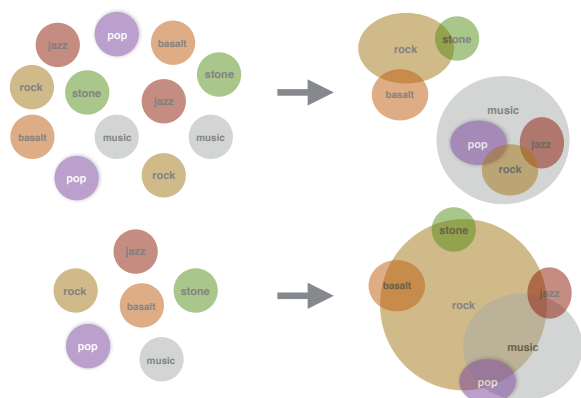

Figure 1: **Top:** Each word is represented by a Gaussian mixture, where each mixture component corresponds to a distinct meaning. Each Gaussian component is represented by an ellipsoid, whose center is specified by the mean vector and contour surface specified by the covariance matrix, reflecting subtleties in meaning and uncertainty. On the left, we show examples of Gaussian mixture distributions of words where Gaussian components are randomly initialized. After training, we see on the right that one component of the word rock is closer to stone and basalt, whereas the other component is closer to jazz and pop. We also demonstrate the entailment concept where the distribution of more general word music encapsulates words such as jazz, rock, pop. **Bottom:** An embedding model where each word is represented by a Gaussian distribution (Vilnis and McCallum, 2014). For words with multiple meanings, such as 'rock', the variance of the learned representation becomes unnecessarily large in order to assign some probability to both meanings. Moreover, the mean vector for such words can be pulled between two clusters, centering the mass of the distribution on a region which is far from certain meanings.

### 3.2 Skip-Gram

The training objective for learning $\theta = \{\vec{\mu}_{w,i}, p_{w,i}, \Sigma_{w,i}\}$ draws inspiration from the continuous skip-gram model (Mikolov et al., 2013a), where word embeddings are trained to maximize the probability of observing a word given another nearby word. This procedure follows the *distributional hypothesis* that words occurring in natural contexts tend to be semantically related. For instance, the words jazz and music tend to occur near one another more often than jazz and cat; hence, jazz and music are more likely to be related. The learned word representation contains useful semantic information and can be used to perform a variety of NLP tasks such as analogy, word similarity analysis, sentiment classification, or as a preprocessed input for complex system such as statistical machine translation.

### 3.3 Energy-based Max-Margin Objective

Each sample in the objective consists of two pairs of words, $(w, c)$ and $(w, c')$. $w$ is sampled from a sentence in a corpus and $c$ is a nearby word within a context window of length $\ell$. For instance, a word $w = $ jazz which occurs in the sentence I listen to jazz music has context words I, listen, to , music. $c'$ is a negative context word (e.g. airplane) obtained from random sampling across the whole document from distribution $P_n(c')$ (details in supplementary material, Section A.1). The objective is to maximize the energy between words that occur near each other, $w$ and $c$, and minimize the energy between $w$ and its negative context $c'$. This approach is similar to negative sampling (Mikolov et al., 2013a,b), which contrasts the dot product between positive context pairs with negative context pairs. The energy function is a measure of similarity between distributions and will be presented and discussed in Section 3.4.

We use a max-margin ranking objective (Joachims, 2002), used for Gaussian embeddings in Vilnis and McCallum (2014), which pushes the similarity of a word and its positive context higher than that of its negative context by a margin $m$:

$$L_\theta(w, c, c') = \max(0,$$
$$m - \log E_\theta(w, c) + \log E_\theta(w, c'))$$

This objective can be minimized by mini-batch stochastic gradient descent with respect to the pa-

rameters $\theta = \{\vec{\mu}_{w,i}, p_{w,i}, \Sigma_{w,i}\}$ – the mean vectors, covariance matrices, and mixture weights – of our multimodal embedding in Eq. (1).

### 3.4 Energy Function

For vector representation of words, a usual choice for similarity measure (energy function) is a dot product between two vectors. Our word representations are distributions instead of point vectors and therefore need a measure that reflects not only the point similarity, but also the uncertainty. In contrast to Vilnis and McCallum (2014), who use a KL energy function, we propose to use the *expected likelihood kernel*, which is a generalization of an inner product between vectors to an inner product between distributions (Jebara et al., 2004). That is,

$$E(f,g) = \int f(x)g(x)\,dx = \langle f, g \rangle_{L_2}$$

where $\langle \cdot, \cdot \rangle_{L_2}$ denotes the inner product in Hilbert space $L_2$. We choose this form of energy since it can be evaluated in a closed form given our choice of probabilistic embedding in Eq. (1).

For Gaussian mixtures $f, g$ representing the words $w_f, w_g$, $f(x) = \sum_{i=1}^{K} p_i \mathcal{N}(x; \vec{\mu}_{f,i}, \Sigma_{f,i})$ and $g(x) = \sum_{i=1}^{K} q_i \mathcal{N}(x; \vec{\mu}_{g,i}, \Sigma_{g,i})$, $\sum_{i=1}^{K} p_i = 1$, and $\sum_{i=1}^{K} q_i = 1$, we find (See Section A.1) the log energy is

$$\log E_\theta(f,g) = \log \sum_{j=1}^{K} \sum_{i=1}^{K} p_i q_j e^{\xi_{i,j}} \quad (2)$$

where

$$\xi_{i,j} \equiv \log \mathcal{N}(0; \vec{\mu}_{f,i} - \vec{\mu}_{g,j}, \Sigma_{f,i} + \Sigma_{g,j})$$
$$= -\frac{1}{2} \log \det(\Sigma_{f,i} + \Sigma_{g,j}) - \frac{D}{2} \log(2\pi)$$
$$-\frac{1}{2}(\vec{\mu}_{f,i} - \vec{\mu}_{g,j})^\top (\Sigma_{f,i} + \Sigma_{g,j})^{-1}(\vec{\mu}_{f,i} - \vec{\mu}_{g,j})$$
$$(3)$$

We call the term $\xi_{i,j}$ partial (log) energy. Observe that this term captures the similarity between the $i^{th}$ meaning of word $w_f$ and the $j^{th}$ meaning of word $w_g$. The total energy in Equation 2 is the sum of possible pairs of partial energies, weighted accordingly by the mixture probabilities $p_i$ and $q_j$.

The term $-(\vec{\mu}_{f,i} - \vec{\mu}_{g,j})^\top (\Sigma_{f,i} + \Sigma_{g,j})^{-1}(\vec{\mu}_{f,i} - \vec{\mu}_{g,j})$ in $\xi_{i,j}$ explains the difference in mean vectors of semantic pair $(w_f, i)$ and $(w_g, j)$. If the semantic uncertainty (covariance) for both pairs are low,

this term has more importance relative to other terms due to the inverse covariance scaling. We observe that the loss function $L_\theta$ in Section 3.3 attains a low value when $E_\theta(w, c)$ is relatively high. This could be achieved by $\vec{\mu}_{f,i}$ being close to $\vec{\mu}_{g,j}$, which brings the term close to zero. It could also be achieved by high values of $\Sigma_{f,i}$ and $\Sigma_{g,j}$, which washes out the importance of the mean vector difference. The term $-\log \det(\Sigma_{f,i} + \Sigma_{g,j})$ serves as a regularizer that prevents the covariances from being pushed too high at the expense of learning a good mean embedding.

At the beginning of training, $\xi_{i,j}$ roughly are on the same scale among all pairs $(i, j)$'s. During this time, all components learn the signals from the word occurrences equally. As training progresses and the semantic representation of each mixture becomes more clear, there can be one term of $\xi_{i,j}$'s that is predominantly higher than other terms, giving rise to a semantic pair that is most related.

The negative KL divergence is another sensible choice of energy function, providing an asymmetric metric between word distributions. However, unlike the expected likelihood kernel, KL divergence does not have a closed form if the two distributions are Gaussian mixtures.

## 4 Experiments

We have introduced a model for multi-prototype embeddings, which expressively captures word meanings with whole probability distributions. We show that our combination of energy and objective functions, proposed in section 3, enables one to learn interpretable multimodal distributions through unsupervised training, for describing words with multiple distinct meanings. Our model also reduces the unnecessarily large variance of a Gaussian embedding model, and has improved results on word entailment tasks.

To learn the parameters of the proposed mixture model, we train on a concatenation of two datasets: UKWAC (2.5 billion tokens) and Wackypedia (1 billion tokens) (Baroni et al., 2009). We discard words that occur fewer than 100 times in the corpus, which results in a vocabulary size of $314,129$ words. Unless stated otherwise, our Gaussian mixture model has two Gaussian components (see Section 4.2 for discussion on higher number of components). Our word sampling scheme is similar to that of word2vec with one negative context word for each positive

| Word | Co. | Nearest Neighbors |
|------|-----|-------------------|
| rock | 0 | basalt:1, boulder:1, boulders:0, stalagmites:0, stalactites:0, rocks:1, sand:0, quartzite:1, bedrock:0 |
| rock | 1 | rock/:1, ska:0, funk:1, pop-rock:1, punk:1, indie-rock:0, band:0, indie:0, pop:1 |
| bank | 0 | banks:1, mouth:1, river:1, River:0, confluence:0, waterway:1, downstream:1, upstream:0, dammed:0 |
| bank | 1 | banks:0, banking:1, banker:0, Banks:1, bankas:1, Citibank:1, Interbank:1, Bankers:0, transactions:1 |
| Apple | 0 | Strawberry:0, Tomato:1, Raspberry:1, Blackberry:1, Apples:0, Pineapple:1, Grape:1, Lemon:0 |
| Apple | 1 | Macintosh:1, Mac:1, OS:1, Amiga:0, Compaq:0, Atari:1, PC:1, Windows:0, iMac:0 |
| star | 0 | stars:0, Quaid:0, starlet:0, Dafoe:0, Stallone:0, Geena:0, Niro:0, Zeta-Jones:1, superstar:0 |
| star | 1 | stars:1, brightest:0, Milky:0, constellation:1, stellar:0, nebula:1, galactic:1, supernova:1, Ophiuchus:1 |
| cell | 0 | cellular:0, Nextel:0, 2-line:0, Sprint:0, phones.:1, pda:1, handset:0, handsets:1, pushbuttons:0 |
| cell | 1 | cytoplasm:0, vesicle:0, cytoplasmic:1, macrophages:0, secreted:1, membrane:0, mitotic:0, endocytosis:1 |
| left | 0 | After:1, back:0, finally:1, eventually:0, broke:0, joined:1, returned:1, after:1, soon:0 |
| left | 1 | right-hand:0, hand:0, right:0, left-hand:0, lefthand:0, arrow:0, turn:0, righthand:0, Left:0 |

| Word | Nearest Neighbors |
|------|-------------------|
| rock | band, bands, Rock, indie, Stones, breakbeat, punk, electronica, funk |
| bank | banks, banking, trader, trading, Bank, capital, Banco, bankers, cash |
| Apple | Macintosh, Microsoft, Windows, Macs, Lite, Intel, Desktop, WordPerfect, Mac |
| star | stars, stellar, brightest, Stars, Galaxy, Stardust, eclipsing, stars., Star |
| cell | cells, DNA, cellular, cytoplasm, membrane, peptide, macrophages, suppressor, vesicles |
| left | leaving, turned, back, then, After, after, immediately, broke, end |

Table 1: Nearest neighbors based on cosine similarity between the mean vectors of Gaussian components for Gaussian mixture embedding (top) and Gaussian embedding (bottom). The notation `w:i` denotes the $i^{th}$ mixture component of the word `w`.

context word. For reproducibility, we describe hyperparameter settings and implementation details in the supplementary material. We performed experiments using 12 cores of Intel i7 3.5GHz CPU. Code will be made publicly available upon publication.

After training, we obtain learned parameters $\{\vec{\mu}_{w,i}, \Sigma_{w,i}, p_i\}_{i=1}^{K}$ for each word $w$. We treat the mean vector $\vec{\mu}_{w,i}$ as the embedding of the $i^{\text{th}}$ mixture component with the covariance matrix $\Sigma_{w,i}$ representing its subtlety and uncertainty. We perform qualitative evaluation to show that our embeddings learn meaningful multi-prototype representations and compare to existing models using a quantitative evaluation on word similarity datasets and word entailment.

We name our model as Word to Gaussian Mixture (`w2gm`) in constrast to Word to Gaussian (`w2g`) (Vilnis and McCallum, 2014). Unless stated otherwise, `w2g` refers to our implementation of `w2gm` model with one mixture component.

## 4.1 Similarity Measures

Since our word embeddings contain multiple vectors and uncertainty parameters per word, we use the following measures that generalizes similarity scores. These measures pick out the component pair with maximum similarity and therefore determine the meanings that are most relevant.

### 4.1.1 Expected Likelihood Kernel

The most natural choice for a similarity score is the expected likelihood kernel, an inner product between distributions, which we discussed in section 3.4. This metric incorporates the uncertainty from the covariance matrices in addition to the similarity between the mean vectors.

### 4.1.2 Maximum Cosine Similarity

This metric measures the maximum similarity of mean vectors among all pairs of mixture components between distributions $f$ and $g$. That is, $d(f,g) = \max\limits_{i,j=1,...,K} \frac{\langle \boldsymbol{\mu}_{f,i}, \boldsymbol{\mu}_{g,j} \rangle}{||\boldsymbol{\mu}_{f,i}|| \cdot ||\boldsymbol{\mu}_{g,j}||}$, which corresponds to matching the meanings of $f$ and $g$ that are the most similar. For a Gaussian embedding, maximum similarity reduces to the usual cosine similarity.

### 4.1.3 Minimum Euclidean Distance

Cosine similarity is popular for evaluating embeddings. However, our training objective directly involves the Euclidean distance in Eq. (3), as opposed to dot product of vectors such as in `word2vec`. Therefore, we also consider the Euclidean metric: $d(f,g) = \min\limits_{i,j=1,...,K} [||\boldsymbol{\mu}_{f,i} - \boldsymbol{\mu}_{g,j}||]$.

## 4.2 Qualitative Evaluation

In Table 1, we show examples of polysemous words and their nearest neighbors in the embed-

ding space to demonstrate that our trained embeddings capture multiple word senses. For instance, a word such as `rock` that could mean either `stone` or `rock music` should have each of its meanings represented by a distinct Gaussian component. Our results for a mixture of two Gaussians model confirm this hypothesis, where we observe that the $0^{th}$ component of `rock` being related to `basalt`, `boulders` and the $1^{st}$ component being related to `indie`, `funk`, `hip-hop`. Similarly, the word `bank` has the $0^{th}$ component representing the river bank and the $1^{st}$ component representing the financial bank.

By contrast, in Table 1 (bottom), see that for Gaussian embeddings with one mixture component, nearest neighbors of polysemous words are predominantly related to a single meaning. For instance, `rock` mostly has neighbors related to rock music and `bank` mostly related to the financial bank. The alternative meanings of these polysemous words are not well represented in the embeddings. As a numerical example, the cosine similarity between `rock` and `stone` for the Gaussian representation of Vilnis and McCallum (2014) is only 0.029, much lower than the cosine similarity 0.586 between the $0^{th}$ component of `rock` and `stone` in our multimodal representation.

**Embedding Visualization**

We provide an anonymous interactive visualization website (`http://35.161.153.223:6002`) for our embeddings (`w2gm`) that allows real-time queries of words' nearest neighbors (in the `embeddings` tab). We use a notation similar to that of Table 1, where a token `w: i` represents the component `i` of a word `w`. For instance, if we search for `bank:0`, we obtain the nearest neighbors such as `river:1`, `confluence:0`, `waterway:1`, which indicates that the $0^{th}$ component of 'bank' has the meaning 'river bank'. On the other hand, searching for `bank:1` yields nearby words such as `banking:1`, `banker:0`, `ATM:0`, indicating that this component is close to the 'financial bank'. We also have a visualization of a unimodal Gaussian embedding (`w2g`) for comparison.

In addition, we include results for our Gaussian mixture model where $K = 3$, which can learn three distinct meanings. For instance, each of the three components of 'cell' is close to ('keypad', 'digits'), ('incarcerated', 'inmate') or ('tissue', 'antibody'), indicating that the distribution

captures the concept of 'cellphone', 'jail cell', or 'biological cell', respectively. We note that due to the limited number of words with more than 2 meanings, our model with $K = 3$ does not generally offer performance differences to our model with $K = 2$; hence, we do not further display $K = 3$ results for compactness.

### 4.3 Word Similarity

We evaluate our embeddings on several standard word similarity datasets, namely, SimLex (Hill et al., 2014), WS or WordSim-353, WS-S (similarity), WS-R (relatedness) (Finkelstein et al., 2002), MEN (Bruni et al., 2014), MC (Miller and Charles, 1991), RG (Rubenstein and Goodenough, 1965), YP (Yang and Powers, 2006), MTurk(-287,-771) (Radinsky et al., 2011; Halawi et al., 2012), and RW (Luong et al.). Each dataset contains a list of word pairs with a human score of how related or similar the two words are.

We calculate the Spearman correlation (Spearman, 1904) between the labels and our scores generated by the embeddings. The Spearman correlation is a rank-based correlation measure that assesses how well the scores describe the true labels.

The correlation results are shown in Table 2 using the scores generated from the expected likelihood kernel, maximum cosine similarity, and maximum Euclidean distance.

We show the results of our Gaussian mixture model and compare the performance with that of `word2vec` and the original Gaussian embedding by Vilnis and McCallum (2014). We note that our model of a unimodal Gaussian embedding `w2g` also outperforms the original model, which differs in model hyperparameters and initialization, for most datasets. Our multi-prototype model `w2gm` also performs better than skip-gram or Gaussian embedding methods on many datasets, namely, `WS`, `WS-R`, `MEN`, `MC`, `RG`, `YP`, `MT-287`, `RW`. The maximum cosine similarity yields the best performance on most datasets; however, the minimum Euclidean distance is a better metric for the datasets `MC` and `RW`. These results are consistent for both the single-prototype and the multi-prototype models.

We also compare out results on WordSim-353 with the multi-prototype embedding method by Huang et al. (2012), shown in Table 3. We observe that our single-prototype model `w2g` is competitive compared to models by Huang et al.

| Dataset | sg* | w2g* | w2g/mc | w2g/el | w2g/me | w2gm/mc | w2gm/el | w2gm/me |
|---------|-----|------|--------|--------|--------|---------|---------|---------|
| SL | 29.39 | **32.23** | 29.35 | 25.44 | 25.43 | 29.31 | 26.02 | 27.59 |
| WS | 59.89 | 65.49 | 71.53 | 61.51 | 64.04 | **73.47** | 62.85 | 66.39 |
| WS-S | 69.86 | 76.15 | 76.70 | 70.57 | 72.3 | **76.73** | 70.08 | 73.3 |
| WS-R | 53.03 | 58.96 | 68.34 | 54.4 | 55.43 | **71.75** | 57.98 | 60.13 |
| MEN | 70.27 | 71.31 | 72.58 | 67.81 | 65.53 | **73.55** | 68.5 | 67.7 |
| MC | 63.96 | 70.41 | 76.48 | 72.70 | **80.66** | 79.08 | 76.75 | 80.33 |
| RG | 70.01 | 71 | 73.30 | 72.29 | 72.12 | **74.51** | 71.55 | 73.52 |
| YP | 39.34 | 41.5 | 41.96 | 38.38 | 36.41 | **45.07** | 39.18 | 38.58 |
| MT-287 | - | - | 64.79 | 57.5 | 58.31 | **66.60** | 57.24 | 60.61 |
| MT-771 | - | - | **60.86** | 55.89 | 54.12 | 60.82 | 57.26 | 56.43 |
| RW | - | - | 28.78 | 32.34 | 33.16 | 28.62 | 31.64 | **35.27** |

Table 2: Spearman correlation for word similarity datasets. The models `sg`, `w2g`, `w2gm` denote `word2vec` skip-gram, Gaussian embedding, and Gaussian mixture embedding (K=2). The measures `mc`, `el`, `me` denote maximum cosine similarity, expected likelihood kernel, and minimum Euclidean distance. For each of `w2g` and `w2gm`, we underline the similarity metric with the best score. For each dataset, we boldface the score with the best performance across all models. We note that the correlation scores for `sg*`, `w2g*` are taken from Vilnis and McCallum (2014) and the scores for `w2g` and `w2mg` are trained with window size 10.

| MODEL | $\rho \times 100$ |
|-------|-------------------|
| HUANG | 64.2 |
| HUANG* | 71.3 |
| W2G | 70.9 |
| W2GM | **73.5** |

Table 3: Spearman's correlation ($\rho$) on WordSim-353 datasets for our Word to Mixture of Gaussians (W2MG) embeddings, as well as the multi-prototype embedding by Huang et al. (2012). `Huang*` is trained using data with all stop words removed. `W2MG-(S,M)` refers to our model with 1 or 2 mixture components.

(2012), even without using a corpus with stop words removed. This could be due to the auto-calibration of importance via the covariance learning which decrease the importance of very frequent words such as `the`, `to`, `a`, etc. Moreover, our multi-prototype model substantially outperforms the model of Huang et al. (2012) on the WordSim-353 dataset.

## 4.4 Word Similarity for Polysemous and Homonymous Words

We use the dataset SCWS introduced by Huang et al. (2012), where word pairs are chosen to have variations in meanings of polysemous and homonymous words.

We compare our method with multiprototype models by `Huang` (Huang et al., 2012), `Tian` (Tian et al., 2014), and `Chen` (Chen et al., 2014).

| MODEL | DIMENSION | $\rho \times 100$ |
|-------|-----------|-------------------|
| WORD2VEC SKIP-GRAM | 50 | 61.7 |
| HUANG-S | 50 | 58.6 |
| W2G | 50 | **64.7** |
| CHEN-S | 200 | 64.2 |
| W2G | 200 | **66.2** |
| HUANG-M AVGSIM | 50 | 62.8 |
| TIAN-M MAXSIM | 50 | **63.6** |
| W2GM MAXSIM | 50 | 62.7 |
| CHEN-M AVGSIM | 200 | **66.2** |
| W2GM MAXSIM | 200 | 65.5 |

Table 4: Spearman's correlation $\rho$ on dataset SCWS. We show the results for single prototype (top) and multi-prototype (bottom) The suffix `-(S,M)` refers to single and multiple prototype models, respectively. Our model `w2g` performs the best among all single-prototype models for either 50 or 200 vector dimensions. Our model `w2gm` performs competitively compared to other multi-prototype models.

We note that `Chen` model uses an external lexical source `WordNet` that gives it an extra advantage.

We use many metrics to calculate the scores for the Spearman correlation. `MaxSim` refers to the maximum cosine similarity. `AveSim` is the average of cosine similarities with respect to the component probabilities.

| MODEL | SCORE | BEST AP | BEST F1 |
|---|---|---|---|
| W2G (5) | COS | 73.1 | 76.4 |
| W2G (5) | KL | 73.7 | 76.0 |
| W2GM (5) | COS | 73.6 | 76.3 |
| W2GM (5) | KL | 75.7 | 77.9 |
| W2G (10) | COS | 73.0 | 76.1 |
| W2G (10) | KL | 74.2 | 76.1 |
| W2GM (10) | COS | 72.9 | 75.6 |
| W2GM (10) | KL | 74.7 | 76.3 |

Table 5: Entailment results for models `w2g` and `w2gm` with window size 5 and 10. The metrics used are the maximum cosine similarity, or the maximum negative KL divergence. We calculate the best average precision as well as the best F1 score. `w2gm` consistently outperforms `w2g` for describing entailment.

## 4.5 Reduction in Variance of Polysemous Words

One motivation for our Gaussian mixture embedding is to model word uncertainty more accurately than Gaussian embeddings, which can have overly large variances for polysemous words. We see that our Gaussian mixture model does indeed reduce the variances of each component for such words. For instance, we observe that the word `rock` in `w2g` also has much higher variance per dimension ($e^{-1.8} \approx 1.65$) compared to that of Gaussian components of `rock` in `w2gm` (which has variance of roughly $e^{-2.5} \approx 0.82$). We also see, in the next section, that the Gaussian mixture model has desirable qualitative behavior for word entailment.

## 4.6 Word Entailment

We evaluate our embeddings on the word entailment dataset from Baroni et al. (2012). The lexical entailment between words is denoted by $w_1 \models w_2$ which means that all instances of $w_1$ are $w_2$. The entailment dataset contains positive pairs such as *aircraft $\models$ vehicle* and negative pairs such as *aircraft $\not\models$ insect*.

We generate entailment scores of word pairs and find the best threshold, measured by Average Precision (AP) or F1 score, which identifies negative versus positive entailment. We use the maximum cosine similarity and the minimum KL divergence, $d(f, g) = \min_{i,j=1,...,K} KL(f||g)$, for entailment scores. The minimum KL divergence is similar to the maximum cosine similarity, but also incorporates the embedding uncertainty. And KL divergence is an asymmetric measure, which is

more suitable for certain tasks such as word entailment. For instance, $w_1 \models w_2$ does not imply $w_2 \models w_1$ such as *aircraft $\models$ vehicle* and *car $\models$ vehicle*. The difference between $KL(w_1||w_2)$ versus $KL(w_2||w_1)$ distinguishes which word distribution encompasses another distribution, a concept demonstrated in Figure 1.

Table 5 shows the results of our `w2gm` model versus the Gaussian embedding model `w2g`. We observe a trend for both models with window size 5 and 10 that the KL metric yields improvement (both AP and F1) over cosine similarity. In addition, `w2gm` has a better performance compared to `w2g`. The multi-prototype model estimates the meaning uncertainty better since it is no longer constrained to be unimodal, leading to better characterizations of entailment. On the other hand, the Gaussian embedding model suffers from large variance problem for polysemous words, which results in less informative word distribution and inferior entailment scores.

## 5 Discussion

We introduced a model that represents words with expressive multimodal distributions formed from Gaussian mixtures. To learn the properties of each mixture, we proposed an analytic energy function for combination with a maximum margin objective. The resulting embeddings capture different semantics of polysemous words, uncertainty, and entailment, and also perform favorably on word similarity benchmarks.

Elsewhere, latent probabilistic representations are proving to be exceptionally valuable, able to capture nuances such as face angles with variational autoencoders (Kingma and Welling, 2013) or subtleties in painting strokes with the InfoGAN (Chen et al., 2016). Similarly, probabilistic word embeddings can capture a range of subtle meanings, and advance the state of the art in predictive tasks. In the future, such representations could also open the doors to a new suite of applications in language modelling, where word distributions are used as inputs to probabilistic LSTMs, or in decision functions where uncertainty matters.

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

# A Supplementary Material

## A.1 Derivation of Expected Likelihood Kernel

We derive the form of expected likelihood kernel for Gaussian mixtures. Let $f, g$ be Gaussian mixture distributions representing the words $w_f, w_g$. That is, $f(x) = \sum_{i=1}^{K} p_i \mathcal{N}(x; \mu_{f,i}, \Sigma_{f,i})$ and $g(x) = \sum_{i=1}^{K} q_i \mathcal{N}(x; \mu_{g,i}, \Sigma_{g,i})$, $\sum_{i=1}^{K} p_i = 1$, and $\sum_{i=1}^{K} q_i = 1$. The expected likelihood kernel is given by

$$
\begin{aligned}
E_\theta(f, g) &= \int \left( \sum_{i=1}^{K} p_i \mathcal{N}(x; \mu_{f,i}, \Sigma_{f,i}) \right) \cdot \\
&\quad \left( \sum_{j=1}^{K} q_j \mathcal{N}(x; \mu_{g,j}, \Sigma_{g,j}) \right) dx \\
&= \sum_{i=1}^{K} \sum_{j=1}^{K} p_i q_j \int \mathcal{N}(x; \mu_{f,i}, \Sigma_{f,i}) \cdot \mathcal{N}(x; \mu_{g,j}, \Sigma_{g,j}) \, dx \\
&= \sum_{i=1}^{K} \sum_{j=1}^{K} p_i q_j \mathcal{N}(0; \mu_{f,i} - \mu_{g,j}, \Sigma_{f,i} + \Sigma_{g,j}) \\
&= \sum_{i=1}^{K} \sum_{j=1}^{K} p_i q_j e^{\xi_{i,j}}
\end{aligned}
$$

where we note that $\int \mathcal{N}(x; \mu_i, \Sigma_i) \mathcal{N}(x; \mu_j, \Sigma_j) \, dx = \mathcal{N}(0, \mu_i - \mu_j, \Sigma_i + \Sigma_j)$ (Vilnis and McCallum, 2014) and $\xi_{i,j}$ is the log partial energy, given by equation 3.

## A.2 Implementation

In this section we discuss practical details for training the proposed model. Our code is implemented with Tensorflow (Abadi et al., 2015) and will be made publicly available upon publication.

### Word Sampling

We use a word sampling scheme similar to the implementation in `word2vec` (Mikolov et al., 2013a,b). We use window size 10, unless stated otherwise, to generate the the context words for a given center word.

Frequent words such as `the`, `a`, `to` are not as meaningful as relatively more infrequent words such as `dog`, `love`, `rock` and we are often more interested in learning the semantics of the less frequently observed words.

We use the subsampling method to balance the importance of frequent words and rare words, which has been shown to improve the performance of learning word vectors (Mikolov et al., 2013b). This technique discards word $w_i$ with probability $P(w_i) = 1 - \sqrt{t/f(w_i)}$, where $f(w_i)$ is the frequency of word $w_i$ in the training corpus and $t$ is a frequency threshold. We use $t = 10^{-5}$ in our experiments, which is the recommended value for `word2vec` skip-gram on large datasets.

To generate negative context words, each word type $w_i$ is sampled according to a distribution $P_n(w_i) \propto U(w_i)^{3/4}$ which is a distorted unigram distribution $U(w_i)^{3/4}$ that also serves to diminish the relative importance of frequent words. Both subsampling and the negative distribution choice are proven effective in `word2vec` training (Mikolov et al., 2013b).

### Reduction to Diagonal Covariance

We use a diagonal $\Sigma$, in which case inverting the covariance matrix is trivial and computations are particularly efficient.

Let $\boldsymbol{d}^f, \boldsymbol{d}^g$ denote the diagonal vectors of $\Sigma_f, \Sigma_g$ The expression for $\xi_{i,j}$ reduces to

$$\xi_{i,j} = -\frac{1}{2}\sum_{r=1}^{D}\log(d_r^p + d_r^q)$$

$$-\frac{1}{2}\sum\left[(\boldsymbol{\mu}_{p,i} - \boldsymbol{\mu}_{q,j}) \circ \frac{1}{\boldsymbol{d}^p + \boldsymbol{d}^q} \circ (\boldsymbol{\mu}_{p,i} - \boldsymbol{\mu}_{q,j})\right]$$

where $\circ$ denotes element-wise multiplication. The spherical case which we use in all our experiments is similar since we simply replace a vector $\boldsymbol{d}$ with a single value.

## Optimization Constraint and Stability

We optimize $\log \boldsymbol{d}$ since each component of diagonal vector $\boldsymbol{d}$ is constrained to be positive. Similarly, we constrain the probability $p_i$ to be in $[0, 1]$ and sum to 1 by optimizing over unconstrained scores $s_i \in (-\infty, \infty)$ and using a softmax function to convert the scores to probability $p_i = \frac{e^{s_i}}{\sum_{j=1}^{K} e^{s_j}}$.

The loss computation can be numerically unstable if elements of the diagonal covariances are very small, due to the term $\log(d_r^f + d_r^g)$ and $\frac{1}{\boldsymbol{d}^q + \boldsymbol{d}^p}$. Therefore, we add a small constant $\epsilon = 10^{-4}$ so that $d_r^f + d_r^g$ and $\boldsymbol{d}^q + \boldsymbol{d}^p$ becomes $d_r^f + d_r^g + \epsilon$ and $\boldsymbol{d}^q + \boldsymbol{d}^p + \epsilon$.

In addition, we observe that $\xi_{i,j}$ can be very small which would result in $e^{\xi_{i,j}} \approx 0$ up to machine precision. In order to stabilize the computation in eq. 2, we compute its equivalent form

$$\log E(f, g) = \xi_{i',j'} + \log \sum_{j=1}^{K}\sum_{i=1}^{K} p_i q_j e^{\xi_{i,j} - \xi_{i',j'}}$$

where $\xi_{i',j'} = \max_{i,j} \xi_{i,j}$.

## Model Parameters and Training Details

In the loss function $L_\theta$, we use a margin $m = 1$. We initialize the word embeddings with $D$ dimension with a uniform distribution over $[-\sqrt{\frac{3}{D}}, \sqrt{\frac{3}{D}}]$ so that the expectation of variance is 1 and the mean is zero. (LeCun et al., 1998) In our experiments, we use the embedding size $D = 50$, unless stated otherwise. We initialize each dimension of the diagonal matrix (or a single value for spherical case) with a constant value $v = 0.05$, which yields a faster convergence given the scale of the norm used above. We also initialize the mixture scores $s_i$ to be 0 so that the initial probabilities are equal among all $K$ components. We use a batch size of 128.

We also use a separate output embeddings in addition to input embeddings, similar to `word2vec` implementation (Mikolov et al., 2013a,b). That is, each word has two sets of distributions $q_I$ and $q_O$, each of which is a Gaussian mixture. For a given pair of word and context $(w, c)$, we use the input distribution $q_I$ for $w$ (input word) and the output distribution $q_O$ for context $c$ (output word). We optimize the parameters of both $q_I$ and $q_O$ and use the trained input distributions $q_I$ as our final word representations.

We use mini-batch asynchronous gradient descent with Adagrad (Duchi et al., 2011) which performs adaptive learning rate for each parameter. We also experiment with Adam (Kingma and Ba, 2014) which corrects the bias in adaptive gradient update of Adagrad and is proven very popular for most recent neural network models. However, we found that it is much slower than Adagrad ($\approx 10$ times). This is because the gradient computation of the model is relatively fast, so a complex gradient update algorithm such as Adam becomes the bottleneck in the optimization. Therefore, we choose to use Adagrad which allows us to better scale to large datasets. We use a linearly decreasing learning rate from 0.05 to 0.00001.

