# Peer review of "Multimodal Word Distributions"

_ACL 2017 — decision unknown_

[Official Review · Reviewer 1 · rating 4 · confidence 5]
soundness 4 · originality 3 · clarity 5 · impact 2 · substance 4 · appropriateness 5 · meaningful comparison 3 · presentation format Oral Presentation

Review: Multimodal Word Distributions

- Strengths:  Overall a very strong paper.

- Weaknesses: The comparison against similar approaches could be extended.

- General Discussion:

The main focus of this paper is the introduction of a new model for learning
multimodal word distributions formed from Gaussian mixtures for multiple word
meanings. i. e. representing a word by a set of many Gaussian distributions. 
The approach, extend the model introduced by Vilnis and McCallum (2014) which
represented word as unimodal Gaussian distribution. By using a multimodal, the
current approach attain the problem of polysemy.

Overall, a very strong paper, well structured and clear. The experimentation is
correct and the qualitative analysis made in table 1 shows results as expected
from the approach.  There’s not much that can be faulted and all my comments
below are meant to help the paper gain additional clarity. 

Some comments: 

_ It may be interesting to include a brief explanation of the differences
between the approach from Tian et al. 2014 and the current one. Both split
single word representation into multiple prototypes by using a mixture model. 

_ There are some missing citations that could me mentioned in related work as :

Efficient Non-parametric Estimation of Multiple Embeddings per Word in Vector
Space Neelakantan, A., Shankar. J. Passos, A., McCallum. EMNLP 2014
Do Multi-Sense Embeddings Improve Natural Language Understanding? Li and
Jurafsky, EMNLP 2015
Topical Word Embeddings. Liu Y., Liu Z., Chua T.,Sun M. AAAI 2015

_ Also, the inclusion of the result from those approaches in tables 3 and 4
could be interesting. 

_ A question to the authors: What do you attribute the loss of performance of
w2gm against w2g in the analysis of SWCS?

I have read the response.

[Official Review · Reviewer 2 · rating 4 · confidence 3]
soundness 4 · originality 3 · clarity 3 · impact 2 · substance 4 · appropriateness 5 · meaningful comparison 3 · presentation format Poster

This work uses Gaussian mixtures to represent words and demonstrates its
potential in capturing multiple word meanings for polysemy. The training
process is done based on a max-margin objective. The expected likelihood kernel
is used as the similarity between two words' distributions. Experiment results
on word similarity and entailment tasks show the effectiveness of the proposed
work.

- Strengths:

The problem is clearly motivated and defined. Gaussian mixtures are much more
expressive than deterministic vector representations. It can potentially
capture different word meanings by its modes, along with probability mass and
uncertainty around those modes. This work represents an important contribution
to word embedding. 

This work propose a max-margin learning objective with closed-form similarity
measurement for efficient training.

This paper is mostly well written. 

- Weaknesses:

See below for some questions. 

- General Discussion:

In the Gaussian mixture models, the number of gaussian components (k) is
usually an important parameter. In the experiments of this paper, k is set to
2. What is your criteria to select k? Does the increase of k hurt the
performance of this model? What does the learned distribution look like for a
word that only has one popular meaning?

I notice that you use the spherical case in all the experiments (the covariance
matrix reduces to a single number). Is this purely for computation efficiency?
I wonder what's the performance of using a general diagonal covariance matrix.
Since in this more general case, the gaussian mixture defines different degrees
of uncertainty along different directions in the semantic space, which seems
more interesting.

Minor comments:
Table 4 is not referred to in the text.
In reference, Luong et al. lacks the publication year.

I have read the response.